# Influencing Factors for Sustainable Dietary Transformation—A Case Study of German Food Consumption

**DOI:** 10.3390/foods11020227

**Published:** 2022-01-15

**Authors:** Nadine Seubelt, Amelie Michalke, Tobias Gaugler

**Affiliations:** 1Faculty of Mathematics, Natural Sciences and Materials Engineering, Institute of Materials Resource Management, University of Augsburg, 86159 Augsburg, Germany; nadine.seubelt@uni-a.de; 2Chair of Applied Geography and Sustainability Sciences, University of Greifswald, 17489 Greifswald, Germany; tobias.gaugler@uni-greifswald.de

**Keywords:** sustainable consumption, dietary behavior, food markets, case study, sustainable transformation

## Abstract

In a case study of Germany, we examine current food consumption along the three pillars of sustainability to evaluate external factors that influence consumers’ dietary decisions. We investigate to what extent diets meet nutritional requirements (social factor), the diets’ environmental impact (ecological factor), and the food prices’ influence on purchasing behavior (economic factor). For this, we compare two dietary recommendations (plant-based, omnivorous) with the status quo, and we examine different consumption styles (conventional, organic produce). Additionally, we evaluate 1446 prices of food items from three store types (organic store, supermarket, and discounter). With this, we are able to evaluate and compare 30 different food baskets along their health, environmental, and economic impact. Results show that purchasing decisions are only slightly influenced by health-related factors. Furthermore, few consumers align their diet with low environmental impact. In contrast, a large share of consumers opt for cheap foods, regardless of health and environmental consequences. We find that price is, arguably, the main factor in food choices from a sustainability standpoint. Action should be taken by policy makers to financially incentivize consumers in favor of healthy and environmentally friendly diets. Otherwise, the status quo further drives especially underprivileged consumers towards unhealthy and environmentally damaging consumption.

## 1. Introduction

Empty supermarket shelves, hoarding, and lack of food and hygiene products, such as pasta, yeast, or toilet paper in grocery stores [1] caused existential fears all over the world at the beginning of the Corona Pandemic. COVID-19 gave the industrialized population, in particular, a small glimpse of what it was like to worry about one’s daily food supply, as was the case in the post-war era. 

At the end of World War II, famine and resource scarcity plagued nations due to low agricultural yields and unstable food security. The top priority was defeating these resource shortages and ensuring stable food security without a focus on healthy and balanced nutrition just yet [2,3]. To reach these goals, the Food and Agriculture Organization (FAO) was founded in 1945 [4]. With the economic boom throughout the 20th-mid-century, fears of food insecurity subsided in the global north and, rather, led to overconsumption. In 1950–1960, for example, consumption of poultry meat tripled per capita per year, and pork consumption also increased from 19 to 30 kilos per capita per year in Germany [3]. As a result, obesity and associated diseases increased sharply [3]. This raised the question of which foods can benefit health and nutrition. As early as 1950, the first dietary guidelines were developed for this purpose, intended to help people align their lifestyles with healthy food choices. These guidelines did not change significantly over time [2]. Later on, the nutrition circle, created by the German Nutrition Society (DGE), was introduced in Germany as a didactic tool [5]. 

Simultaneously, while food intake and, thus, calorie intake increased physical labor and with it energy demand decreased with advancing technology. For example, the proportion of employees working office jobs steadily rises [6]. The heightened prosperity and a wide range of food choices and social pressure to opt for convenience foods rather than healthy options, increases the number of people suffering from malnutrition or overweight [7]. Scientists and institutes have been warning of the health risks of increased food consumption for years. With this, the supposed health industry seems to be booming [8], with sales of diet products increasing over 30% since 2013 [9]. Alongside this, the trend towards a meat-free diet is growing. In the last 6 years, the number of vegans in Germany has already increased by 33% [10]. Research also shows that a vegan diet brings health benefits and reduces diseases such as diabetes and cardiovascular disease [11,12,13]. More and more people are open to a healthy lifestyle based on a healthy and balanced diet [14], yet there are reasons (sociocultural, emotional, etc.), which can hold them back in doing so [15,16,17,18].

Against this background, the first research question arises: namely, to what extent the population in industrialized countries actually eats a healthy diet? 

The scarcity of resources during and after World War II led policy makers and researchers to develop new innovations and advances in the food industry. As a policy tool for this, the Common Agricultural Policy (CAP) was introduced in the European Economic Community to increase food supply and facilitate access to it. The CAP has had a significant impact on food supply, food prices, and the environment. However, it has also had a major impact on the way food is produced [2]. The use of inorganic fertilizers, herbicides, and pesticides was necessary to achieve the required productivity. However, the attempt to avoid crop failure, increase yields, and thus, combat hunger has been accompanied by the exploitation and destruction of the earth. 

Excessive agriculture consumes enormous amounts of water and land, endangered ecosystems, and causes a large amount of greenhouse gas emissions [4,19,20]. Nowadays, the food system is responsible for 15–31% of total greenhouse gas emissions in Europe [4,5]. In Germany, the share is estimated at 15–20% [5,21], which means about 1.7 tons of greenhouse gas emissions per capita. 

In addition to the conventional farming methods known today, organic farming developed, aiming at environmentally friendly agriculture. Organic farming helps build soil fertility, maintain biodiversity, and reduce losses of nitrogen, phosphorus, and pesticides [22,23]. Although it is arguable if greenhouse gas pollution is lower compared to conventional agriculture [24], the environmental benefits, in terms of ecosystem services that organic farms provide, are an absolute good [22,25]. The trend towards environmentally conscious diets has been on the rise since the 2000s. This can be seen in the increasing sales of organic products: they have doubled in Germany since 2011 and are already at 14.99 million euros in 2020 [26]. Additionally, the demand for organically traded products can be seen in the growing number of organic farms. Currently, every 8th farm in Germany represents a form of organic farming, and already, 10.2% of agricultural land in Germany is farmed according to organic guidelines [27].

Alongside this, environmental awareness is reflected in changing eating habits of the population and environmental sustainability metrics have been identified as an important pillar in nutrition education [28]. As mentioned, the number of vegans, i.e., people who abstain from all products of animal origin, and vegetarians, who largely abstain from meat, is growing [29]. This is because it is precisely the production of animal foods that causes a significant proportion of environmental damage with its high demand for resources but low production efficiency [30]. Abstaining from eating animal foods would be an important step to reduce climate damage [13,19,24,31,32,33]. For example, a vegan diet causes 40% less carbon dioxide emissions and would cause an average of only 1040 kg of CO2 emissions per capita in Germany [34].

As sustainable development of basically all sectors is of rising importance and sustainable consumption of all goods is necessary [35], knowledge about the strong environmental impact of food is growing [36,37,38,39], and different population strata behave more or less sustainably in response to this [16]. The question that arises is whether knowledge is put to action and to what extent the population in industrialized countries does eat environmentally consciously? 

New developments in the cultivation and production of food also impact its price. During the famine at the end of World War II, for example, the cost of food was still immense and large proportion of consumption expenditure was spent on food and beverages [40]. However, with the increasing economic growth, the situation turned, and within 100 years, the proportionate expenditures in Germany for food and beverages sank from 57% in 1900 to only 15% in 2000 [41]. Prices for food products have decreased in recent years due to better use of fertilizers and technology, and in turn, the average income has increased due to economic growth in industrialized countries. In fact, the higher the GDP, the lower the share of spending on food [42]]. Compared to other European countries, Germany is far below the average of consumer spending [40]. A reason for this could be economic incentives. Across all media outlets, the cheapest offers from grocery stores are advertised. Discounters, in particular, have been dueling for years with the lowest price promise, conveying to the consumer that food has to be cheap. Especially for people with low incomes, the price represents a decisive driver [43]. However, it’s not just financially underprivileged people who are cutting back on the quality of their food. For example, in a qualitative survey in the UK, the cost of food was cited as the most common reason for eating unhealthily across all income groups [44]. 

Against the background of economic factors addressed here, the third and final research question arises: to what extent does the German population base their purchase decision on the price of food?

In summary, it is the aim of this study to identify the influence on consumers’ food choices from three different aspects: diets are analyzed along the lines of (1) nutritional health, (2) impacts on the environment, as well as (3) market situation or food prices. On this basis it is discussed whether social, ecological, or economic factors impact consumption choices and to what extent qualitative guidelines or the market situation hold potential of improving consumers’ dietary behavior. 

The paper starts with a description of the methodology and the data. Following, results and findings are presented and discussed for a conclusion.

## 2. Materials and Methods

In the following, the methodology for determining social, ecological, and economic influences on food consumption are addressed. The method is used in a case study within the German context. However, it can be transferred to comparable geopolitical frames, in particular to other highly developed countries in the western world.

### 2.1. Goal and Scope Definition

First, national dietary recommendations defined by the DGE and the Giessen Vegan Food Pyramid (GVFP) are compared to current dietary habits for insight into a potential disparity between supposedly healthy eating and actually consumed diets. Next, current eating habits and purchasing behaviors are analyzed to paint a picture on the present level of environmentally conscious food consumption in Germany. The current average dietary habit is defined below as the status quo. Further dietary styles considered in this assessment are an omnivore (defined by the DGE) and a plant-based diet (defined by the GVFP). This is further combined with two different forms of production practice, namely conventional or organic production. The combination yields four types of purchasing styles: omnivorous and conventional, omnivorous and organic, plant-based and conventional, and lastly, plant-based and organic. Third, economic implications for consumers are examined. For this, the dietary status quo, as well as shopping baskets, defined based on previously mentioned purchasing styles, are evaluated regarding the foods’ prices. Three different price levels for each product category are also investigated by tracing the actual market prices charged in three different types of grocery stores. This is done to depict consumers’ varying financial means underlying their purchasing behavior. Resulting is a definition of 30 in the following so-called shopping baskets.

### 2.2. Methods and Data for the Social Consideration

To compare health aspects of food choices, a comparison is made between the current average German diet and two dietary recommendations of German associations, which represent an omnivorous (DGE) as well as a plant-based diet (GVFP). By considering a plant-based dietary recommendation, the status quo is compared with this allegedly more ecologically sustainable alternative. At the same time, a balanced plant-based diet is widely established as healthy due to the lack of consumption of meat and other animal products [11]. In fact, the consumption of meat, in particular, is associated with the risk of higher mortality, cardiovascular disease, and certain forms of cancer [12], which is why this comparison is used for the consideration of health effects caused by different dietary styles.

The current diet is presented on the basis of the annual per capita consumption of various foods, determined by the BMEL [45]. The dietary recommendations are based on the recommendations of the DGE for an omnivorous diet [46], as well as the GVFP [47] for a plant-based diet. These guidelines give quantities for food intake that guarantee sufficient supply of essential nutrients. The weight ranges given for each food were averaged to an accurate serving size for the average person. The diets are based on the following food categories: “Grain and Cereal Products”, “Vegetable and Pulses”, “Fruit and Nuts”, “Milk and Milk products”, “Milk and Milk product alternatives”, “Eggs (shell weight)”, “Meat, Sausage and Fish”, “Additional Food”, “Fats” and “Sugar” [45].

### 2.3. Methods and Data for the Ecological Consideration

As a second factor of influence on consumers’ food choices, environmental awareness is analyzed. Different products have varying impact on the environment. Thus, a plant-based diet causes a much smaller ecological footprint than an omnivorous diet and can be considered an overall sustainable alternative [48,49,50]. Similarly, organic agricultural production does less damage to the environment than conventional processes, for example through the use of fewer pesticides and respect for biodiversity. It is, thus, broadly considered the more sustainable practice [2,25]. Therefore, the proportions of organic foods currently purchased in Germany are examined to determine the influence of the environmental factor on the consumers’ choice of food. This survey is provided by the Verbrauchs und Medienanalyse (VuMA) [51,52]. Furthermore, the proportions of different diets within the German population—omnivorous, vegetarian, and vegan—are analyzed to draw conclusions about the extent to which ecological awareness already impacts eating habits. In this context, survey results are obtained by the Allensbacher Markt und Werbeträgeranalyse (AWA) [10]. In addition, the attitude of German consumers towards social and ecological responsibility is considered and is compared with the two previous surveys in order to relate the current ecological attitude of Germans to their consumer behavior. The survey data is also provided by VuMa [53]. 

### 2.4. Methods and Data for the Social Consideration

In order to identify how food prices influence consumers’ dietary behavior, market research is carried out for the German food market. With this, current prices of groceries were determined. Therefore, different shopping baskets were created, as described previously, which contain a defined selection of products: The shopping baskets are based both on the previously mentioned omnivorous and plant-based dietary recommendation. Both recommendations contain amounts of the individual food groups within a certain range (which, depending on the food, is given e.g., in grams or pieces). These amounts also account for the necessary nutrient supply of one healthy adult. The weight ranges given for each food are averaged to provide an accurate serving size for the average person. Since the recommendations do not provide more detailed differentiation on the selection of specific fruits, vegetables, and meats, these were determined by the average per capita consumption of Germans [45,54] to represent current consumption decisions accurately. The amount of food is determined per one week and one person to provide good comparability. Table 1 presents the final shopping list, based on the nutritional recommendations of DGE (left column) and GVFP (right column).

The prices of the foods within those baskets were determined with a market analysis. For this, three different types of food stores were considered to portray the German food sector fairly accurately, as they offer groceries at different price levels. The stores considered are (a) a full range supermarket, (b) a discounter, and (c) an organic food store. In this case study, (a) is a REWE market, representing a large chain of 33,000 stores distributed throughout Germany; (b) is represented by the discounter LIDL, which operates 10,800 outlets in 32 countries; (c) is ebl-naturkost, a small-scale organic food store, with 30 branches located in Bavaria in the South of Germany [56]. 

Since a distinction was made between organic quality and conventional production, the latter is not found in (c) the organic market; prices for conventional products were hence only collected in stores (a) and (b). 

There are several alternative products for the same food (e.g., a no-name product/ private label/ branded product). Prices within the predefined shopping baskets were collected for the cheapest, a middle-priced, and the most expensive offers within each store to depict the price dispersion within supermarkets. If less than three different price levels were available for one product, the lowest price was used to fill the gaps. 

When products were only available in organic quality (even in stores (a) and (b)), prices for conventional products were taken as the available organic price. Even after supplementing some in-store unavailable product prices with prices listed within the stores’ online shops, 10% of prices were still unavailable. The organic assortment was particularly small for the supermarket (a), and the discounter (b). These remaining missing prices were established on the basis of the average deviation between the organic store’s (c) and the respective missing store’s prices. A detailed description of the procedure, based on an example, can be found in the Appendix B.

To ensure comparability, the prices of the 61 products were collected over a period of only three weeks in spring 2021. They were collected as prices per kg, with the product size closest to the full kg selected for the market analysis. Finally, the total price of the shopping basket was calculated according to the identified prices per kilo and the respective dietary recommendations defining the baskets. This market analysis results in a total of 30 shopping baskets (Figure 1) and in a total survey size of 1446 prices.

### 2.5. Uncertainty

Due to market, seasonal, and regional fluctuations, all prices collected are subject to a certain degree of inaccuracy. This is largely irrelevant for our market analysis since price volatility is taken into account to some extent: random price fluctuations would have an impact across all markets and would not reverse the final results and implications. Further, a wide variety of products is considered, which helps compensate for any extrema that might be occurring at the time the market was analyzed. However, seasonal fluctuations in market prices, or even in the general products’ supply, are not considered. Shopping baskets further represent, as already mentioned, examples for the average German adult. Depending on one’s individual preferences or habits, this is not representative for every citizen, but it is rather used for explanations and general. Further, the described calculation of missing prices represents only an approximation of the prices. However, this only affects a minor proportion of the prices surveyed (12%) and, otherwise, a large number of products could not have been included in the evaluation. Generally, it is arguable if only a comparison of plant-based vs. omnivorous and organic vs. conventional production is a sufficient metric to determine sustainability of different diets. There are other components to be considered in the context of food sustainability. However, to draw general conclusions, we decided to define this as an approximation to a sustainability metric for this paper.

## 3. Results

In the following, results from considering societal, environmental, as well as economic influencing factors on peoples’ dietary choices are presented. Section 3.1 describes how health recommendations are comparable to the status quo. Furthermore, a closer look at current ecological performance of dietary specifications is given in Section 3.2. Finally, the price of all described dietary types is examined as an influencing factor on consumption behavior with the focus of results on the market analysis, presented in Section 3.3.

### 3.1. Social Consideration

Figure 2 shows the comparison of current dietary consumption and the recommendations of the DGE [30,46] and GVFP [47].

For all three cases, the four main food sources are cereal products, fruits, vegetables, and milk (products) or alternatives. It is apparent that the dietary status quo in Germany deviates from the health recommendations in many areas. The current average diet consists of 57% plant-based foods (excluding sugar and fats). The DGE recommends almost double the intake of vegetables and pulses (38%), resulting in a plant-based share of 66%. According to the GVFP, this share should even increase to almost three quarters (73%) of the total diet. 

The proportion of milk and dairy products or their alternatives are rather comparable within the three diets, with 18.6% (status quo), 21.8% (DGE), and 22.2% (GVFP). The consumption of primary animal-based products, such as meat (products) and fish, is much higher than recommended with 0.705 kg more than described as the maximum intake by the DGE. The consumption of eggs and fats in Germany is currently also higher than recommended by both nutrition guidelines. In addition, sugar is consumed as 7% of the overall average diet, whereas it is completely excluded in both dietary recommendations. 

The lack of consumption of nutrient dense foods, such as vegetables or pulses, especially, indicate an unbalanced prevailing diet amongst the German population.

### 3.2. Ecological Consideration

Subsequently, the current dietary consumption in Germany is analyzed regarding its ecological performance and whether this indicates an influence on consumers’ dietary behavior. 

Diet has a strong impact on the environment. High meat consumption is responsible for a significant amount of greenhouse gases, as well as water consumption [13,31,48,58]. Similarly, it is known that conventional farming causes higher damage to the environment compared to organic production [22,24,25]. Therefore, transitioning towards a plant-based and organic diet would be a valuable step in contributing to a healthy environment and fighting climate change [58].

Figure 3 shows the current proportions of diets, the share of German consumers buying organic foodstuff, and their attitudes towards socially and environmentally responsible products.

Figure 3 shows that only 38.2% of surveyed Germans regularly purchase organic products. The larger part, in contrast, states to rarely or never buy organic foods [29]. Despite the fact that the trend of meat-free diets has been increasing in recent years, this group still makes up no more than 9.2% of the total population of Germany [29]. Only 1.4% of Germans consume a vegan diet, which is considered most sustainable compared to omnivorous or vegetarian diets [10,50]. However, comparing these actual purchasing decisions with the consumers’ statements on the importance of socially and ecologically produced products indicates a significant attitude-behavior gap: over half of surveyed people state their personal interest in a sustainably responsible way of producing as fully or mostly true. This gap has been shown by other studies likewise [15,16]. Even though social and ecological responsibility as a purchasing criterion has increased in recent years [52,53], this does not yet have a pertinent effect on German consumption behavior in buying organically grown products as a sustainable form of diet.

### 3.3. Economic Consideration

In this section, the results of the market research are analyzed as they are compared with the current average expenditure for food in Germany. The average expenditure of a German consumer is largely similar to the prices of the cheapest examined price level [59]. Therefore, only this price group is considered in detail below. The results of the remaining price levels can be found in the Appendix A.

First, the price differences of the four purchasing styles—based upon the described dietary recommendations and agricultural production practices—are considered: plant-based (GVFP) and organic, plant-based and conventional, omnivorous (DGE) and organic, omnivorous and conventional, as well as the status quo of German food consumption. For this comparison, an average is calculated from the three store types considered. We find that, on average, a plant-based diet is 15% more expensive than an omnivorous diet. An organic purchase averages to almost double the price (+99%) than an otherwise identical basket of conventional products. Looking at Table 2, the price difference between omnivore and plant-based diets is larger when purchasing conventional products (+41%) than when opting for organic foods, where a plant-based diet is only slightly more expensive (+3%) than its omnivorous pendant. What is apparent, however, is the greater expense, when opting for the most sustainable shopping style, i.e., plant-based and organic: it is more than twice as expensive (+144%) than the more environmentally damaging conventional omnivorous shopping style.

The results in Figure 4 show that a diet based on the recommendations—either plant-based from DVFP, with an average of 43.26 €, or omnivorous from DGE, with an average of 36.02 €—is well within the average expenditure on food among Germans (44 € on average). However, it is also clear that the average consumer would need to invest at least 15% more for healthy and environmentally sound procurement.

When purchasing conventional products only, a healthy, and partly sustainable (plant-based) diet can be afforded well within current expenditure for food. However, if sustainable production practices (organic) are to be taken into consideration as well, a 6.59 € (omnivorous) or 8.21 € (plant-based) price increase per week is expected compared to current expenses. This amounts to about 343 €, or about 427 € per year for an omnivorous or vegan diet, respectively. For one average household (1.99 capita), this would mean about 683 €, or 850 € of additional expenses per year. In both cases, this is more than twice a monthly grocery budget and represents rather large additional costs.

To work out the differences between the purchasing decisions in more detail, a look is taken at the food groups and cultivation forms within the different dietary styles, as well as the current average expenses of German consumers (Table 2 and Figure 5).

At first glance, organic meat products, as well as organic vegetables and pulses, make up the most expensive food groups. This leads to a similar price for omnivorous or plant-based diets when purchasing organically.

The difference between the plant-based and omnivorous diet for plant-based foods overall is striking, as the expenditure for fruit within a plant-based diet is twice as high as within an omnivorous diet. This is reasonable, considering that, according to the dietary recommendation for plant-based nutrition, this diet requires almost twice the amount of fruit as the omnivorous recommendation suggests. The amount of vegetables is also 67% higher within the plant-based recommended diet.

Germans consume twice the amount of meat that the DGE recommends as the maximum. However, the current average expenditure for meat is just over half the cost it would be if meat were bought in organic quality and in quantity recommended by the DGE. Similarly, spending on fruit and vegetables of almost all purchasing styles is below the minimum cost needed within a diet covering nutritional recommendations. This result is consistent with the finding that German average fruit and vegetable consumption is currently below the dietary recommendations. In addition, spending on other items such as sweets, alcohol, and tobacco is particularly high, at almost 10 €, and represents the highest price share within the status quo. This expenditure is not covered in any of the dietary recommendations and hence increases the cost of current dietary behavior. 

In the following, the different types of cultivation are highlighted. Table 3 shows the average prices of the different food groups for both conventional and organic production. It also shows the percentage deviation of the organic price to the conventional price. Looking at Table 3, causes of the high price difference in the omnivorous diet become apparent: The organic group of meat, sausage, fish, and eggs is by far the most expensive group. In comparison, the price of conventional meat, sausage, fish, and egg is far below at only 30% of the organic price. A very small difference, however, is visible within the group of dairy product alternatives. Since the stores’ house brands are often produced in organic quality, the price for such alternatives is quite low within the organic group (Appendix A). In addition, conventional plant-based food alternatives are oftentimes brand products, which are higher priced generally and thus create a balance between the organic and conventional product prices.

At last, the price differences between the three grocery stores are discussed. As can be seen in Figure 6, a conventional purchase, based on an omnivorous diet, is cheapest in the supermarket at 20.98 €. This is surprising, since shopping at a discounter would be anticipated to yield the lowest prices. However, with a maximum difference of 14% (between discounter and organic store in category omnivorous and organic), the three stores are at similar price levels in the individual dietary and purchasing styles. If one decides to buy organic quality, it makes little difference in the supermarket whether they consume a plant-based or omnivorous diet; in the organic store, a plant-based purchase even performs better than an omnivorous diet, which may be due to the high meat prices. At the discounter, however, it presents as rather the opposite to this. An organic purchase in the organic store also does not necessarily have to be the most expensive; a plant-based organic diet purchased in the supermarket is more expensive.

Table A4 in Appendix C provides a more detailed overview of the prices within the different purchasing styles for each grocery store and food category. In addition, it contains the information on the current average expenditure of a German consumer (status quo). Firstly, it shows that expenditure for omnivorous and conventional products, from both supermarket and discounter, are similarly high to the current average expenditure, while the organic expenditure turns out to be more expensive generally. In the cereal and meat product categories, larger price differences between the current average spending, and omnivore and conventional prices can be observed for the supermarket and the discounter. Thus, the average expenditures in these categories are significantly higher than the required expenses for consuming a nutritionally sound diet. 

Figure 7 takes a closer look at the differences between animal products from the individual stores. It is noticeable that organic milk is, at most, half as expensive as its conventional pendant. It also shows that the price in the organic store is the highest in most cases. The organic store purchases most animal products from regional farms, which might be a reason for the higher prices. The cheapest conventional meat is sold by the discounter; the supermarket is cheaper for organic fish and sausage, however. Poultry and pork tend to be cheaper than beef over all grocery stores. 

Figure 7 takes a closer look at the differences between the animal products of the individual stores. It is noticeable that organic milk is, at most, double as expensive as its conventional pendant. It also shows that prices in the organic store are the highest in most cases. The cheapest conventional meat is sold by the discounter; the supermarket is cheaper for organic fish and sausage, however. Poultry and pork tend to be cheaper than beef over all grocery stores.

## 4. Discussion

Firstly, we find a clear deviation of current shopping behavior from dietary recommendations. It is reasonable to assume that the examined factor—one’s own health—plays a minor role in food selection. The average German diet deviates from the recommendations, notably, in an excessive consumption of fat, sugar, and meat products. Dietary guidelines are currently available for 90 countries [61], which, as shown in 3.1, fail to motivate consumers to follow healthy eating habits. This could be because an individual’s food choice is influenced by a multitude of indicators: biological reasons (e.g., intolerances), social factors (e.g., food-related traditions, social identity, awareness, economic situation) [62,63,64,65], or a constant exposure to external cues to food (e.g., easy access high calorie foods, diet-related media) [66] have shown to complicate healthy eating endeavors. 

The increasing number of vegans suggests that awareness about the diets’ influence on the environment paves the way for environmentally sustainable dietary transitions, but the number of plant-based eaters is too low to establish the environmental factor of significant influence on the choice of food. Further, our results show that the growing, yet too low interest in ecologically produced food also supports this assumption. There is an attitude behavior gap, as even though more than half of German citizens want to buy environmentally sensible products, only about one third show this attentiveness by buying at least some organic products. Accordingly, various reasons must lead to this large discrepancy and prevent people from consuming in an environmentally conscious way. O’Riordan and Stoll-Kleemann (2015) [15], for example, conclude that one barrier to shifting diets towards more plant-based consumption is the aversion of policy makers or practitioners (e.g., food retail) to promote such kind of behavior as it is delicate to communicate to voters or risking economic profit for companies. It is also important to consider that peoples’ emotions, or sociocultural factors, can hold them back from consuming less animal-based foods [18].

This transitions well to the third part of this study: assessing the economic factor along its influence on food choices. The fact that the price of food has a strong influence on food choices is consistent with a qualitative study in the UK by Puddephatt et al. [44]. Our results indicate that the generally higher costs of plant-based and organic products seem to be an important reason for the rather unhealthy, environmentally unfriendly status quo and why an omnivorous, conventional shopping basket is preferred by the average German customer. Because organic products cost, on average, twice as much as conventionally produced ones, and a plant-based diet on average is 15% more expensive than an omnivorous diet, it is clear that there is no financial incentive given for buying more sustainably. Further, if one chooses to follow an omnivorous diet, an economic incentive is not set regarding organically produced meat: they are faced with more than double the cost of current meat expenditure, when following the DGE recommendation, which even suggests lower meat consumption than the status quo, and purchasing organic meat. Hence, one must be able to afford a sustainable purchase. On average, Germans spend 14% of their income on food, beverages, and tobacco [60]. The absolute spendable sum can be restrictive for people with lower incomes. Thus, factoring in sustainability when purchasing food is likely to be a luxury decision.

However, it is observed that the expenditure for a healthy diet, based on both the recommendations of the DGE (21.44 €) and the GVFP (30.30 €), is lower than the average German expenditure on food and beverages per week (44 €). Although these prices do not apply to organic quality, it is still possible to consume a healthy diet at a reasonable cost in Germany. Results show that currently, German consumers spend too small a share of their food expenses on fruit and vegetables. A predominantly environmentally sustainable diet can also be obtained, rather inexpensively, on the basis of the GVFP. This is in line with Macdiarmind et al. (2012) [67] who also find that a nutritious diet, which reduces impacts to climate compared to the status quo, can be consumed without raising costs for the consumer. Additionally, it can be seen that 14% of the actual food expenses alone are attributable to the consumption of sugar and confectionery, alcoholic beverages, and tobacco currently. What is interesting, is that these costs hold the highest share in food expenses overall (over 10 €); this economic weight does not seem to drive consumers away from such consumption habits. By reducing the level of consumption of these foods, customers could save money to invest in more environmentally friendly and health-conscious food alternatives. 

When looking at the results regarding the different types of retail stores, it is evident that organic food is not necessarily more expensive in an exclusively organic market; fruit, vegetables or even cereal products are cheaper in the organic market or at a similar price level compared to the discounter and the supermarket. In addition, the organic market offers a larger selection of organic products. It also offers more regional and unpackaged products, which makes the purchase even more sustainable.

Interestingly, the cheapest shopping basket is not offered by the discounter, as would be expected. An omnivorous and conventional basket purchased in the supermarket induces the fewest expenses for customers. This is due to the higher costs for vegetables and fruits in the discounter, which are offered at a smaller price in the supermarket. Fruits and vegetables that have a particularly high share, such as tomatoes, apples, or grapes, are cheaper at the supermarket. However, these prices for fruit and vegetables are especially subject to seasonal price fluctuations and can differ when assessing the shopping baskets at different points in time. For all other food baskets, however, the discounter provides the cheapest option for consumers. 

## 5. Conclusions

This paper set out to analyze three different possible influencing factors on consumers’ dietary decisions. This work first provides some perspective on the overlap of sustainably preferable dietary patterns—concerning health and environmental favorability—and the actual consumption habits of the German population. It focuses, however, on the correlation between foods’ prices and amounts purchased. This gives insight on shortcomings of the current food market and whether it is designed to support holistically sustainable food consumption. It is groundwork for further research in the context of dietary transitions, and it can function as food for thought for policy makers.

This work shows that the currently prevailing diet of the average German customer is not quite at the nourishing level that renowned dietary recommendations suggest. It is debatable whether more educational campaigns will help foster a transition towards healthier dietary patterns. It could be that yet more information within the context “health and diet” will overwhelm consumers with an already oversaturated market of ever changing “diet wisdom”. What remains clear, however, is that insufficient consumption of fruit and vegetables contrasts, exceeding consumption of sugar and fats within the average German diet, as is the prevailing case in developed countries generally. This should be taken seriously when aiming at campaigns for healthy consumption and also in regulations of food marketing, which oftentimes advertises for unhealthy, highly processed products. In addition, nutrition education can help to develop appropriate educational strategies to achieve healthy eating behavior [68].

There is a trend towards more environmentally conscious diets amongst German consumers. However, an attitude-behavior-gap shows between consumers claiming to be invested in environmentally sound products and their actual lacking consumption of such. Moreover, this eco-conscious trend cannot yet contradict the detriment to the environment caused by production practices that have been established throughout the intensification of agriculture in recent decades. Again, the impact of informational campaigning alone is debatable. However, raising peoples’ knowledge of the food-environment context will definitely not hinder sustainable dietary transition and should, likewise, be fostered by policy makers.

Both on health and environmental level, further research should investigate motivation and willingness to change from different consumer strata. This will provide information on how to best foster dietary transitions for policy-makers and practitioners alike.

Since information on diet alone seemingly has no sufficient effect on a sustainable transition of consumption patterns, the cost of groceries might influence dietary decisions. Our results show that low prices of unsustainable options, as is the case for conventional meat for example, are reflected in high consumption levels of these categories. Further, the market analysis showed that both a plant-based and omnivorous shopping basket with exclusively organic products exceeds current spending for the average German diet. This might explain the described lack of organic purchases contrary to the interest shown by consumers: the higher price of organic food is a burden many are not willing or able to overcome even if environmental impacts could be reduced. It should be a priority for policy-makers to redirect food production towards more sustainable practices and incentivize a transition towards, e.g., organic production.

Results show, however, that a nutritionally adequate diet, and even a more sustainable plant-based diet, can be purchased for lower expenses than is currently spent for the average diet. This suggests that knowledge of dietary contexts and adequate pricing can be overpowered by external factors, such as marketing for unhealthy alternatives or social pressure to partake in certain consumption. 

Leaving all responsibility for a sustainable transformation of the food sector with the customer seems insufficient. Therefore, policymakers need to build upon this momentum. The already accelerated trend towards healthier, environmentally sensible dietary patterns should be fostered with adequate economic incentives: beneficial effects—or a lack of external costs—should be represented within the products’ prices likewise. Increasing the price of unhealthy and environmentally harmful food, whilst subsidizing healthy and environmentally friendly food, could change the current price structure. The political goal should be sustainable food as the cheapest option for the consumer. This would also be desirable because financially underprivileged parts of the population would no longer be economically compelled to consume unsustainable diets.

Although the price structure of food sectors in other countries, especially in the allegedly developed world, seems comparable to data collected for Germany in this paper, food in Germany is comparatively cheap. This makes it difficult to transfer the herein presented results to other countries. Against this background, the aim of further research should be international market analyses and subsequent comparison of the country-specific results. Further, although we were able to describe a correlation between, e.g., low prices with high consumption of certain products, conclusions on causality are limited. This should be fostered in further research to find how price elasticities influence consumption behavior in detail. While data for food prices was selected from different price levels, the analysis on current consumption patterns do not differentiate between certain population strata. There might be differences in dietary behavior regarding socio-demographic factors, which will also be an interesting approach for further research. Based on this, investigating best practices for transforming dietary trends towards health and ecological sustainability, considering the circumstances of society, seems a sensible research trajectory.

## Figures and Tables

**Figure 1 foods-11-00227-f001:**
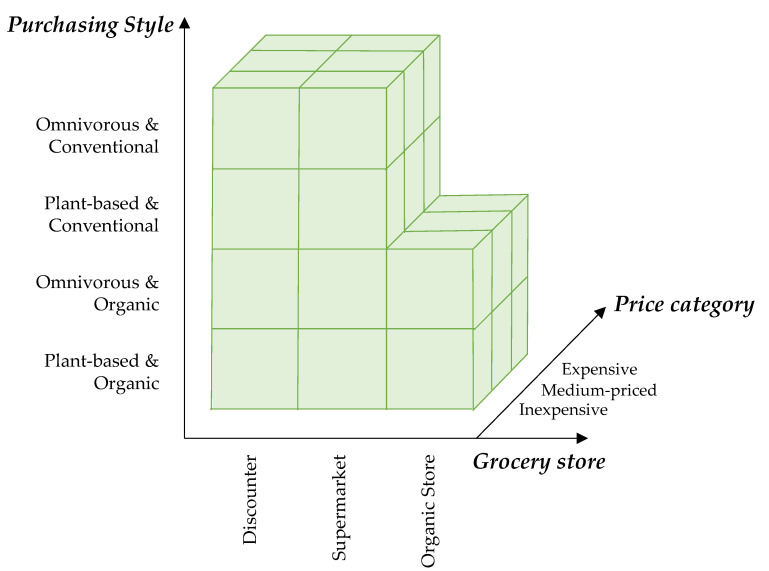
Pictorial representation of the 30 different shopping baskets. The three grocery stores are plotted on the x-axis. In each grocery store, an inexpensive, medium-priced, and expensive product was selected, which is plotted on the z-axis. On the y-axis, the four purchasing styles—omnivorous and conventional, omnivorous and organic, plant-based and conventional, and lastly, plant-based and organic. The organic store does not carry conventional products, so there are no corresponding shopping baskets for this intersection.

**Figure 2 foods-11-00227-f002:**
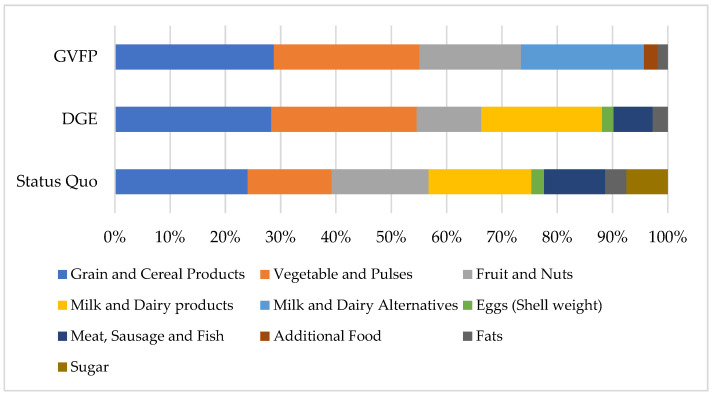
Relative per capita consumption in the status quo [45], and relative consumption recommendations of the DGE [46] and GVFP [47,57].

**Figure 3 foods-11-00227-f003:**
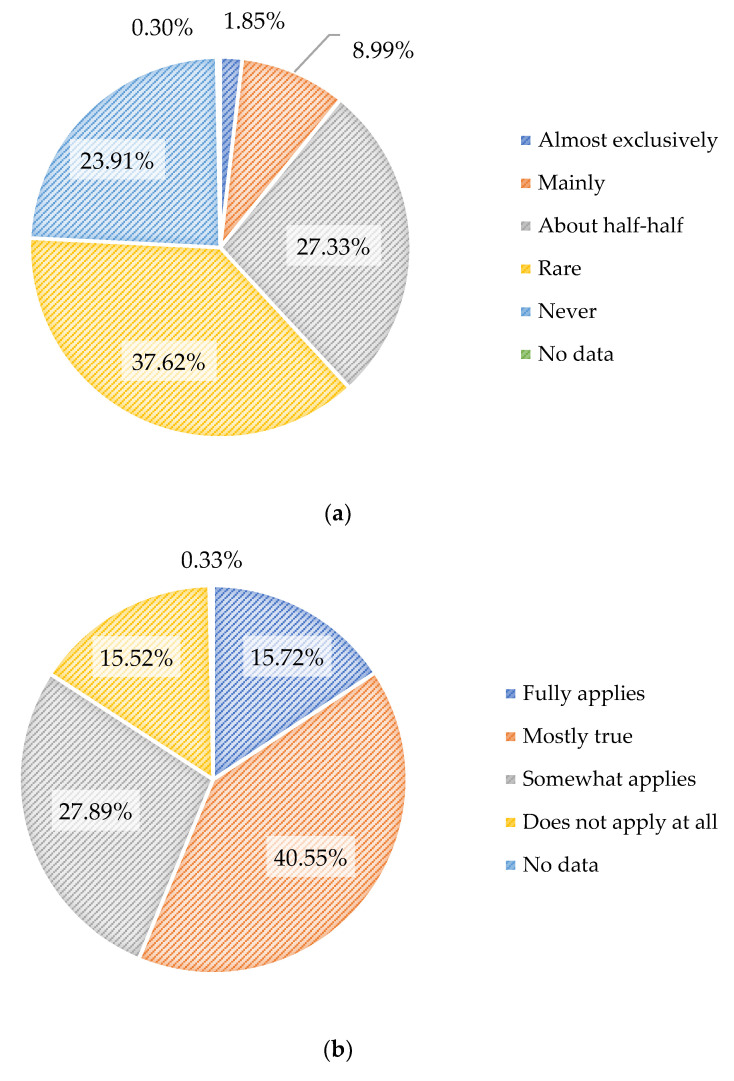
(**a**) Population in Germany by extent of purchase of organic products [51] (**b**) Population in Germany by attitude toward the statement “When I buy products, it is important to me that the respective company acts in a socially and ecologically responsible. manner” [52,53] (**c**) Proportion of vegans and vegetarians in the total population of Germany in 2020 [29].

**Figure 4 foods-11-00227-f004:**
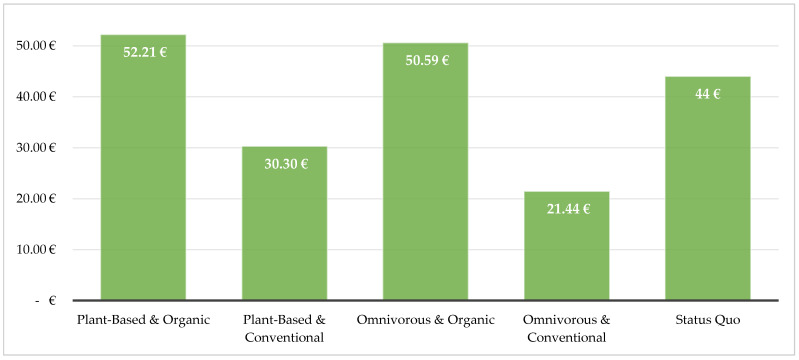
Costs of shopping baskets as an average of all stores, as well as the current average expenditure of Germans on food and beverages (Status Quo).

**Figure 5 foods-11-00227-f005:**
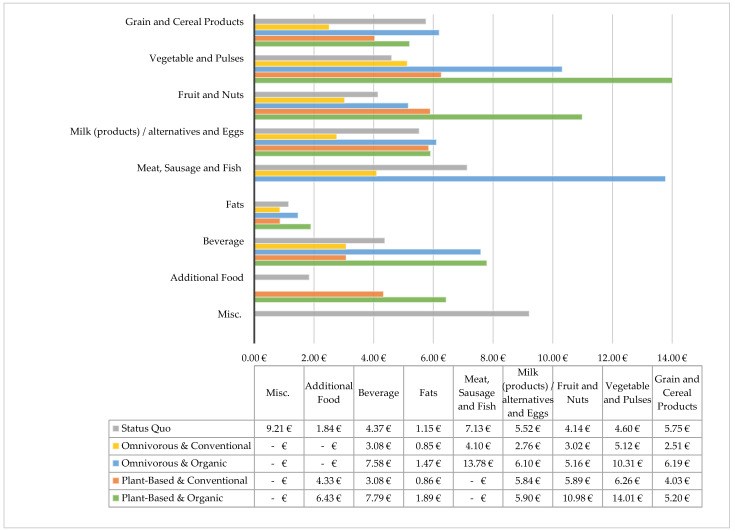
Average prices of the dietary styles separated into different food groups, as well as the current average expenses in Germany [59]. Misc. = miscellaneous; 1 Sugar and confectionery, alcoholic beverage, and Tobacco [60].

**Figure 6 foods-11-00227-f006:**
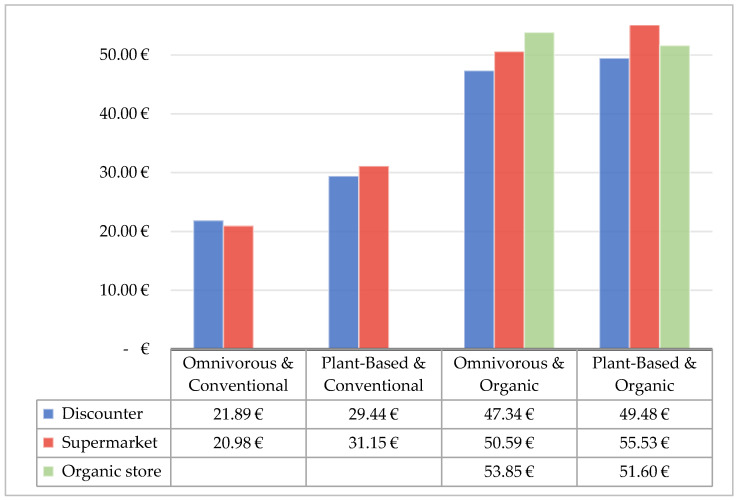
Prices for the shopping baskets for each grocery store.

**Figure 7 foods-11-00227-f007:**
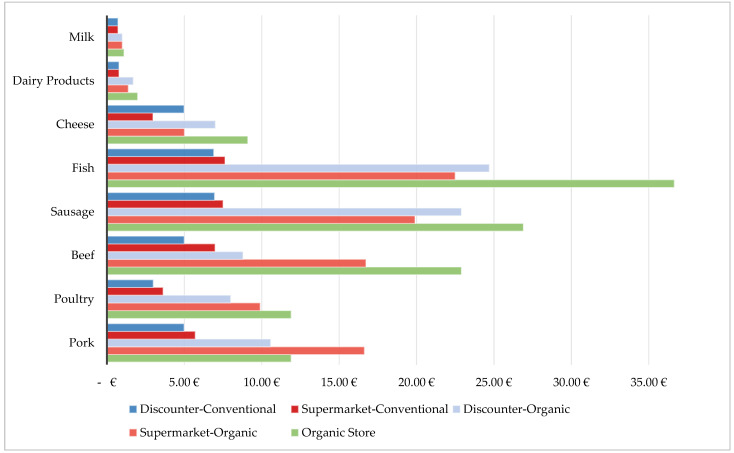
Prices per kilogram for the animal products for each grocery shop.

**Table 1 foods-11-00227-t001:** Shopping-list for an omnivorous diet (left column) and a plant-based diet (right column) calculated for one week and one person.

	Omnivorous Diet (DGE) [g/Week × Person]		Plant-Based Diet (GVFP) [g/Week × Person]	
Grain and Cereal Products, Potatoes	Bread	1575	Wholemeal Bread	656
	Cereal Flakes	193		
	Potatos	525	Potatos	1500
	Noodles	132	Wholemeal Noodles	725
	Rice	116	Rice ^(1)^	355
Vegetables and Pulses ^(3,7)^	Tomatos	680	Tomatos	1020
	Carrots, Red Beet	237	Carrots, Red Beet	355
	Onions	201	Onions	301
	Cucumber	166	Cucumber	250
	Lettuce ^(2)^	138	Lettuce ^(2)^	207
	White/ Red Cabbage	89	White/Red Cabbage	134
	Savoy, Kohlrabi, Chinese Cabbage	55	Savoy, Kohlrabi, Chinese Cabbage	83
	Beans	47	Beans	71
	Mushroom	47	Mushroom	70
	Cauliflower, Green Cabbage, Broccoli	47	Cauliflower, Green Cabbage, Broccoli	70
	Asparagus	43	Asparagus	64
	Spinach	33	Spinach	49
	Peas	30	Peas	45
	Leek	25	Leek	37
	Celery	22	Celery	33
	Brussels Sprout	8	Brussels Sprout	12
	Pulses	490	Pulses	158
Fruit and Nuts ^(7)^	Apple	346	Appel	692
	Banana	187	Banana	375
	Grapes	81	Grapes	161
	Strawberry	60	Strawberry	121
	Peach	58	Peach	116
	Pear	39	Pear	78
	Cherry	38	Cherry	77
	Rasberry	18	Rasberry	35
	Blueberry	16	Blueberry	32
	Plums, Mirabelle	16	Plums, Mirabelle	31
	Apricot	13	Apricot	26
	Blackberry	3	Blackberry	7
	Nuts	175	Nuts	315
Milk and Dairy Products or Alternatives	Milk	787	Soy-, Grain-, Nutdrink	1225
	Yoghurt, Quark, Kefir, Buttermilk	787	Yoghurt-Alternative	1225
	Cheese	385		
Meat, Sausage, Fish and Eggs ^(7)^	Pork	126	/	
	Poultry	61		
	Beef	38		
	Sausage	225		
	Fish, low-fat	115		
	Fish, rich in fat	70		
	Egg	3 pieces		
Oil and Fat	Oil	88	Oil	126
	Butter	79	Linseed Oil	84
	Magarine	79		
Beverage	Water, High-Calcium	3500	Water, High-Calcium	3500
	Non-Alcoholic, Low-Energy Drink ^(4)^	3500	Non-Alcoholic, Low-Energy Drink ^(4)^	3500
	Coffee ^(5)^	228	Coffee ^(5)^	228
Addition	/		Nori	14
			Vitamin B- Supplements	n/a
			Tofu, Seitan, Lupins	263

The following additional assumptions were made in the selection of foods: (1) unlike within the DGE recommendation, unprocessed cereals were not considered here. This is because, on the one hand, rice represents the most important category within this group, and on the other hand, cereals are already represented with the category of bread. —(2) Differently from the data source [45], lettuce was not divided into the two categories “butterhead lettuce/iceberg lettuce” and “other lettuce” but was considered within one category, since supply of the different types of lettuce was not guaranteed in every store. —(3) Currants are not considered because they are only available seasonally and within a short time frame. —(4) Low-energy beverages are assumed to contain less than 10 kcal per 100 milliliters. —(5) One liter of coffee is assumed to require 65 g of coffee powder [55]. —(6) Contrary to the GVFP, tofu, seitan, and lupins are included in the group “in addition” to ensure comparability to DGE within the group “vegetables and pulses”. —(7) “Other fruits”, “other vegetables”, and “other meats”, which the BMEL additionally categorizes, were not considered, as they hold only a small share within the quantities of the individual groups. —(n/a).

**Table 2 foods-11-00227-t002:** Costs of shopping baskets as an average of all stores.

Production Style/Dietary Style	Conventional [per Week × Person]	Organic [per Week × Person]	Difference Organic to Conventional	Average
Omnivorous	21.44 €	50.59 €	+136%	36.02 €
Plant-based	30.30 €	52.21 €	+72%	43.26 €
Difference plant-based to omnivorous	+41%	+3%	-	+15%
Average Price	25.87 €	51.50 €	+99%	38.64 €

**Table 3 foods-11-00227-t003:** Average price per kilo of food groups from all stores, divided into organic and conventional production. Presented is only the cheapest price category.

Groups/Average	Conventional [per kg]	Organic [per kg]	Difference Organic to Conventional
Grain and Cereal Products, Potatoes	1.49 €	2.50 €	+68%
Vegetables and Pulses	2.06 €	4.57 €	+122%
Fruits and Nutzs	5.25 €	7.08 €	+35%
Milk and Dairy Products	0.75 €	1.36 €	+82%
Meat, Sausage, Fish and Eggs	5.22 €	17.63 €	+238%
Oils and Fats	4.80 €	12.50 €	+161%

## Data Availability

Data is contained within the article or Appendix A.

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
