# Peer review of "Influencing Factors for Sustainable Dietary Transformation—A Case Study of German Food Consumption"

_foods, 2022, doi:10.3390/foods11020227_

Round 1
Reviewer 1 Report
REVIEW foods-1515604-peer-review-v1
Dear authors, while this paper is quite interesting, it needs some changes.
1. One major concern which I consider drawback of this article is that continuously you send us to see some section. For example, lines:
223-224: described in section 2.2.
258: As explained in section 2.3
308: as described in 2.4 and Appendix A1
316: is given in section 3.2
318: in section 3.3. Since sections 3.1 and 3.2 are comparisons
471: Table B1 in Appendix B provides
497: Based on the results in section 3.1,
512: The results of section 3.2
So, we are going back and forth trying to figure out what is going on. Additionally, with so many sections it seems like it was a part of a thesis.
The paper is quite lengthy, and this does not help the reader. So, either cut the paper in two papers or see how you will deal with the go to section part
2. Line 161: For this purpose, first, national dietary recommendations are compared..: since you haven’t sated the purpose in the paragraph delete the “for this purpose” since it doesn’t make sense
3. Line 211: different types of diet = omnivorous (?). why do we have =?
4. Line 305: Of course, depending. Delete “of course”. Nothing is “of course” except if it is backed up with references.
5. Line 456: table 4. There is no table 3. From table 2 we go to table 4.
6. Line 499: =one's own health = plays: again why = (?) see if something went wrong in the pdf version.
7. Line 534. Same as 499 lines
8. Provide with your theoretical contribution to academia
9. Extend practical implications
10. Provide with limitations and based on these with directions for further research
I hope that these comments will help you with your work
Best of luck with your paper
Stay healthy and be safe!
Reviewer 2 Report
The paper tackles an interesting and actual topic, about factors for sustainable dietary transformation. I have read the article with careful attention and the following reflections come to my mind
Autor write:
- In 1950-1960, consumption of poultry meat tripled, and pork consumption also increased dramatically [3]. - Question: On what scale? worldwide? national?
- However, while food intake increased, physical labor decreased with advancing technology and new innovations designed to make life easier at work, as well as in every day life. Question: In what sense? Quantity? Calorie content?
- However, the supposedly healthy food industry seems to be booming [8], 64 with sales of diet products increasing over 30% since 2013 [9] (….). In this part the question is: Due to which market developments? Please refer to the global transformation and the growing importance of the sustainable development challenge, where sustainable consumption of different goods is a major component. I recommend Bjelle, E.L.; Wiebe, K.S.; Többen, J.; Tisserant, A.; Ivanova, D.; Vita, G.; Wood, R. Future changes in consumption: The incomeeffect on greenhouse gas emissions. Energy Econ. 2021, 95, 105114.
- More and more people are open to a healthy lifestyle based on a healthy and balanced diet [14]. In this part the reflection is: This, among other reasons, is also due to the growing wastage of food and the attention focused on a balanced diet. This has been written about by the FAO (2010), Gussow and Clancy, (1986), Macdiarmid et al. (2012)
Look:
FAO (2010). Report International Scientifi c Symposium: Biodiversity and Sustainable Diets – United Against Hunger. Rome: FAO Headquarters. ;
Gussow, J., Clancy, K. (1986). Dietary guidelines for sustainability. J. Nutr. Educ., 18, 1–5.;
Macdiarmid, J. I., Kyle, J., Horgan, G. H., Loe, J., Fyfe, C., Johnstone, A., McNeill, G. (2012). Sustainable diets for the future: can we contribute to reducing greenhouse gas emissions by eating a healthy diet? Am. J. Clin. Nutr., 96, 632–639.;
Gazdecki, M.; GoryÅ„ska-Goldmann, E.; Kiss, M.; Szakály, Z. Segmentation of Food Consumers Based on Their Sustainable Attitude. Energies 2021, 14, 3179] ;
- Alongside this, environmental awareness is reflected in the changing eating habits of the population. The number of vegans, i.e., people who abstain from all products of animal origin, has risen as already mentioned and the number of vegetarians, who largely abstain from meat, is also growing [25]. In this part the reflection is: What about those who are afraid to stop eating meat because, for example, they enjoy eating meat very much. Reducing the volume of meat consumed in a day, week or year, switching to vegetarianism is associated by some with a loss. What do you think?
- It is therefore all the more valuable to be able to observe this positive trend towards a vegetarian or even vegan diet. In this part the reflection is: What is needed is a further change in consumer attitudes and behaviour, a change of an ecological nature, meaning that food consumers will turn to plant-based foods and organic food, that they will choose them with awareness, out of interest for the planet, the fight against climate change and a willingness to learn how to live a healthy life. What do the author know about Livewell Plate?
- The higher the GDP, the lower the share of consumption for food. Countries that stand out here are the USA, Great Britain, or Germany [35]. In this part the reflection is: Spending on food?
- I the end of part … For example, in a qualitative 140 survey in the UK, the cost of food was cited as the most common reason for eating un-141 healthily [38]. Reflection: The determining factors in the buying process for many new food products vary depending on the type of innovation and its market acceptance. The authors wrote about this, example Barrena R., Sánchez M., (2013). Neophobia, personal consumer values and novel food acceptance, Food Quality and Preference, Volume 27, Issue 1, Pages 72-84, ISSN 0950-3293, https://doi.org/10.1016/j.foodqual.2012.06.007.
- I the end of part … For this purpose, first, national dietary recommendations a (..) - Question: By whom?
- I the end of part … To compare health aspects of food choices, a comparison is made between the current 181 average German diet and two dietary recommendations of German associations, (….) - - Question: Which ones?
- I the end of part … Therefore, transitioning towards a plant based and organic diet would be a valuable step in contributing to a healthy environment and fighting climate change [54]. (..) - Reflection: This is also important in the context of demographic change, the increase in the number of elderly people.
- I the end of part … However, comparing these actual purchasing decisions with the consumers’ statements on the importance of socially and ecologically produced products indicates a significant attitude-behavior gap: .. (….) - Reflection: This issue is highlighted in the paper - literature review - Gazdecki et al. (2021) Segmentation of Food Consumers Based on Their Sustainable Attitude. Energies 2021, 14, 3179.
- I the end of part … This could be, because an individual’s food choice is influenced by a multitude of indicators: biological reasons (e.g. intolerances), social factors (e.g. food-related traditions, social identity, exposure to diet-related media) [58–60] .. (….) - Reflection: There may be conditions: environmental, economic, those related to consumer awareness, lifestyle, values. There is a lot of work showing the gap between recommendations and consumer behaviour. Care should be taken to review the current literature in greater depth and to confront the results obtained with studies by other authors.
- I the end of part … This paper set out to analyze three different possible influencing factors on consumers’ dietary decisions: the health impacts of different diets, their specific burden on the environment and the associated grocery prices…(….) - Reflection: Unfortunate phrase, does not reflect exactly what the authors have done.
- I the end of part … What remains clear, however, is that insufficient consumption of fruit 578 and vegetables contrasts exceeding consumption of sugar and fats within the average Ger-579 man diet, as is the prevailing case in developed countries generally. …(….) - Reflection: Definitely nutrition education should be provided. Look: Contento I. R. (2008): Nutrition Education: Linking Research, Theory & Practice. Asia Pac J Clin Nutr.;17 Suppl 1:176-9. PMID: 18296331.
- Conclusions - Reflection: An in-depth survey should be planned among the German public on their knowledge of sustainable food consumption issues. This information would make it possible to determine whether conscious, ecological consumers, as compared to others, are motivated by what factors in the choice of food during purchase are important to them. We need research to determine the willingness to change to a more sustainable diet that benefits the health of the consumer. Changing to a more sustainable diet is a long-term process which requires popularisation and re-evaluation of food choice factors, especially of a psychological and sociological nature, as well as consumer attitudes and opinions.
Moreover, the analysis process is thoroughly explained and even if this conception was new to me, as a reviewer, it was easy for me to understand the process and to follow the logic of the paper. At the same the results are well explained and discussed.
In conclusion, the only comments are those listed above.
Overall, this paper belongs to an very interesting stream of literature. Please take the comments more as potential windows to improve the work rather than critics. Good luck.
Round 2
Reviewer 1 Report
dear authors while most of the changes were made please see lines 471 & 474 & 484 where it says table 4 while the previous table is table 2 (line 424)
while line 444 says table 3
put your tables in order - see them again and what they contain
also again (it is probably during conversion so I don't know if the word will have it), 708/709 line there is again "="
also table A2 line 703: cell: "store" something again seems to be in this cell
Reviewer 2 Report
Dear Authors,
thank you for considering my suggestions and I appreciate your response. The presented version of the article is more correct. After reviewing the content of the new Foods - manuscript 1515604-‘Influencing Factors for Sustainable Dietary Transformation – A 2 Case Study of German Food Consumption' I would like to conclude that the issue addressed is in line with the theme of the journal. The problem presented in the article is important and interesting, the research in this area is needed.
The structure of the publication is appropriate, the goals are clearly defined. The methodology used is good, correctly described and supported by literature. The conclusions of the analysis presented are correct and in line with the stated objectives. More literature has been added. Relevant literature has been included in the study, although this part could still be supplemented.
line 157 - monetary factors => suggestion economic factors
line 250 or 251 - Table 1. Shopping… - or 265 => Where is the title? The same question for a Figure 1
line 345 - 2.5 Social Consideration - Numbering errors?
line 348 - if you write the title under the table, the title in the table is unnecessary
line 424 - do you know the MDPI requirements? Do they not indicate that table titles should be before the table?
line 431 - Figure 4 and 6. => ranges created every 10 Euros. The first range is not clearly described in figure 4
Figure 5 and 7 => is it necessary to use capital letters. None of the previous tables and figures practiced this
There are a discussion with references to other studies. Summary: more text has been added.
Good luck with your next research!
